# ANALYZING LOCAL REPRESENTATIONS OF SELF-SUPERVISED VISION TRANSFORMERS

## ABSTRACT

In this paper, we present a comparative analysis of various self-supervised Vision Transformers (ViTs), focusing on their local representative power. Inspired by large language models, we examine the abilities of ViTs to perform various computer vision tasks with little to no fine-tuning. We design evaluation framework to analyze the quality of local, i.e. patch-level, representations in the context of few-shot semantic segmentation, instance identification, object retrieval and tracking. We discover that contrastive learning based methods like DINO produce more universal patch representations that can be immediately applied for downstream tasks with no parameter tuning, compared to masked image modeling. The embeddings learned using the latter approach, e.g. in masked autoencoders, have high variance features that harm distance-based algorithms, such as k-NN, and do not contain useful information for most downstream tasks. Finally, we find an object instance retrieval setting where DINOv2, a model pretrained on two orders of magnitude more data, performs worse than its less compute intensive counterpart DINO.

## 1 INTRODUCTION

Recent advances in Natural Language Processing gave birth to universal models that after large-scale pretraining can perform various language-related tasks without task-specific fine-tuning. Large language models (Brown et al., 2020; OpenAI, 2023; Garcia et al., 2023) based on self-supervised transformers achieve competitive performance on tasks like translation, question answering, and commonsense reasoning with prompting or by in-context learning with just a few examples.

Self-supervised transformers are also getting increasingly popular in computer vision. Two radically different self-supervised learning paradigms have demonstrated good performance for Vision Transformers (ViTs): those based on contrastive learning (e.g. MOCO (He et al., 2020) or DINO (Caron et al., 2021)), and those based on masked image modeling (e.g. MAE (He et al., 2022) or Sim-MIM(Xie et al., 2022)). The question of whether these models possess universal capabilities, similar to those seen in NLP models, for computer vision tasks remains unanswered.

As ViTs do not have text inputs, it is non-trivial to assess their zero-shot capabilities for downstream tasks. Most ViTs produce one embedding vector for the entire image (usually the [CLS] token) and one embedding for each local patch. In this paper we focus on few-shot capabilities of ViTs for vision tasks that require locality awareness, like image segmentation and object tracking. We propose few-shot evaluation methods that leverage patch representations. To minimize task-specific parameter tuning we use two approaches: k-nearest-neighbors with no tuned parameters and linear probing with a single layer of trainable parameters. The power of global image representations from pretrained ViTs for image-level tasks like image classification are relatively well explored in literature, e.g. Caron et al. (2021).

Fine-tuning the entire backbone in addition to relatively large task-specific "heads" still gives superior performance in segmentation and tracking. The analysis of such models is beyond the scope of this paper as their good performance is not just from self-supervised pretraining, but also strongly affected by the head architecture and the data used for fine-tuning. The focus of this work is on the inherent capabilities of self-supervised ViTs that can be exposed by using just a few labeled samples.

We show that while masked image modeling produces backbones with good fine-tuning performance, the frozen, pretrained patch embeddings are far inferior to the ones learned by contrastive methods

for nearest neighbor methods. We dive deep into this phenomenon and identify roughly 200 dataset-agnostic features in the embedding space that, counterintuitely, contain no useful information for the downstream tasks we have considered, while having the highest variances among all features. Removing those features improves k-NN performance for most tasks.

We further explore the usefulness of patch embeddings for identifying the same object instance in multiple images. We perform experiments on a satellite imagery dataset under several image transformations, and find that DINO surprisingly outperforms its newer and larger sibling DINOv2 (Oquab et al., 2023). Additionally, we measure the quality of patch embeddings in distinguishing fine-grained object categories. Lastly, we perform experiments for object association on multi-object tracking datasets. We find that DINO and DINOv2 substantially outperform both masked image models and supervised ViTs, making them most suitable for object retrieval in videos. Our main contributions are summarized as follows:

- We design an evaluation framework along with few-shot datasets to analyze inherent power of pretrained Visual Transformers for locality aware tasks. We analyze and compare five representative ViTs using our framework on three tasks: patch classification, instance and fine-grained retrieval, and object association in video frames.

- We show that compared to masked image modeling, contrastive pretraining produces significantly more universal patch embeddings that can be immediately utilized in downstream tasks without fine-tuning. We identify the cause of poor performance of MAE-like models in methods based on k-NN.

- We show that DINOv2, trained on two orders of magnitude more unlabeled data, outperforms all other ViTs in most scenarios, including in robustness of patch classification with respect to image corruptions. Surprisingly, it underperforms most ViTs in identifying the patches covering the same object instance in transformed images, indicating that blindly adding more data to pretraining might not universally improve all results.

## 2 RELATED WORK

The recent advent of Visual Transformers (ViT) Dosovitskiy et al. (2021); Caron et al. (2021); Oquab et al. (2023) and its use in many downstream tasks has paved the way for novel approaches in several directions of computer vision, including image segmentation Cheng et al. (2022), image classification Dosovitskiy et al. (2021), and object detection Cheng et al. (2022). Unlike Language Models, where the model sizes reached to 175B parameters, scaling ViTs is notoriously difficult Dehghani et al. (2023). Both works DINOv2 Oquab et al. (2023) and ViT-22B Dehghani et al. (2023) claim that their core technical contribution is about stabilizing the training of large transformers on hundreds of millions of images.

Park & Kim (2022) analyze and demonstrate several properties of multi-head self-attentions (MSA) and ViTs. They show that MSAs flatten the loss landscape to alleviate the issue of its non-convexity. They also observe that MSAs and convolutional layers complement each other, showing that MSAs function as low-pass filters, while convolutional layers act as high-pass filters. Park et al. (2023) analyze the differences between the ViT methods that are based on contrastive learning (CL) and masked image modeling (MIM) and compare their performance of downstream tasks. They demonstrate that CL captures longer-range global patterns, such as object shapes, more effectively than MIM methods. Secondly, they demonstrate that CL-based approaches are more shape-oriented, whereas MIM-based approaches are more texture-oriented. Raghu et al. (2021) provide an analysis of the internal representation structure of ViTs and CNNs on several image classification benchmarks. They demonstrate that ViTs have a more uniform representation across the layers of the network compared to CNNs. These differences are mostly explained by the role of self-attention, which allows early aggregation of information, and ViT residual connections, which also propagate features from lower to higher levels.

Other works have focused on analyzing the robustness of ViTs. Bhojanapalli et al. (2021) investigate the robustness of ViT models to input and model perturbations for image classification. Bhojanapalli et al. (2021) demonstrate that transformers are robust to the removal of almost any single layer and that when pre-trained on a sufficiently large dataset, ViTs demonstrate not worse results than the ResNet counterparts across various perturbations. Paul & Chen (2022) analyze the robustness of

ViTs against several common corruptions, perturbations, distribution shifts, and natural adversarial examples. They also analyze and demonstrate the superior robustness of ViTs in various aspects, such as masking, energy/loss landscape analysis, and sensitivity to high-frequency artifacts on robust classification datasets. Naseer et al. (2021) investigate the robustness of Transformers to severe occlusions, perturbations, and domain shifts in classification tasks. Their findings demonstrate that ViTs exhibit significantly less bias towards local textures compared to CNNs.

Another line of research, that contributes to the universality of the models and enables zero-shot image classification (and potentially other vision tasks) involves vision-language models, including contrastive models like CLIP (Radford et al., 2021) and autoregressive models like CM3Leon (Yu et al., 2023). Analysis of these models is beyond the scope of this paper.

Our work analyzes and compares different ViTs regarding their ability to locally represent images. We explore and compare if the local patch representations obtained from ViTs trained with different self-supervised or supervised strategies. To this end, we probe the quality of the patch-wise features for dense patch classification, fine-grained retrieval, and tracking, in a few-shot setting.

## 3 CAN TRANSFORMERS RECOGNIZE SEMANTICS OF PATCHES?

Throughout the paper we use five ViT models. MAE (He et al., 2022) and SimMIM (Xie et al., 2022) are used as representative models of masked image modeling. Contrastive models are represented by DINO (Caron et al., 2021) and its counterpart DINOv2 (Oquab et al., 2023) which unlike all other ViTs used in this work is pretrained on a much larger dataset than ImageNet. We use Supervised ViT (Dosovitskiy et al., 2021) as a baseline, and in one setting we also use iBOT (Zhou et al., 2021) which is trained on ImageNet like DINO but uses loss terms more similar to DINOv2. These models are described in detail in appendix A.1.

To analyze the local representations of the ViT models, we first study their ability to perform patch-wise classification. To this end, we set up a few-shot patch classification experiment on the Cityscapes dataset Cordts et al. (2016).

**Few-shot subset of Cityscapes.** Cityscapes (Cordts et al., 2016) training set consists of 2975 pictures taken in 18 cities. Unless otherwise stated, we use a training dataset consisting of 4 images per city (72 images in total). We later also explore the impact of the number of training sampled by increasing or decreasing this number. The original validation set of the Cityscapes dataset has 500 images, containing images from 3 different cities. For our analyses, we take 10 images per city, resulting in a total of 30 images. We convert the pixel-dense segmentation labels of the Cityscapes dataset to patch-level classes by selecting the most common class within each patch. We evaluate the quality of representations by measuring pixel accuracy and segmentation mIoU.

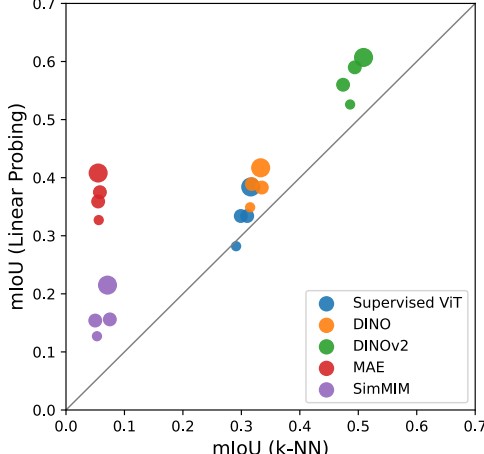

Figure 1: mIoU of patch classifiers on various subsets of Cityscapes dataset. The evaluation is performed on the same validation set. The size of the circles are proportional to the number of images in the training set (36 images, 72 images and 144 images).

As the ViT models used in our analyses take as an input an image of size $224 \times 224$, we tile the images of size $1024 \times 2048$ into $256 \times 256$ tiles and treat each tile as a separate image. The tiles are resized to $224 \times 224$ and passed to the pre-trained transformers. We extract and store the patch representations of the corresponding transformer for all the patches in all the images (training and validation) and for all the ViTs.

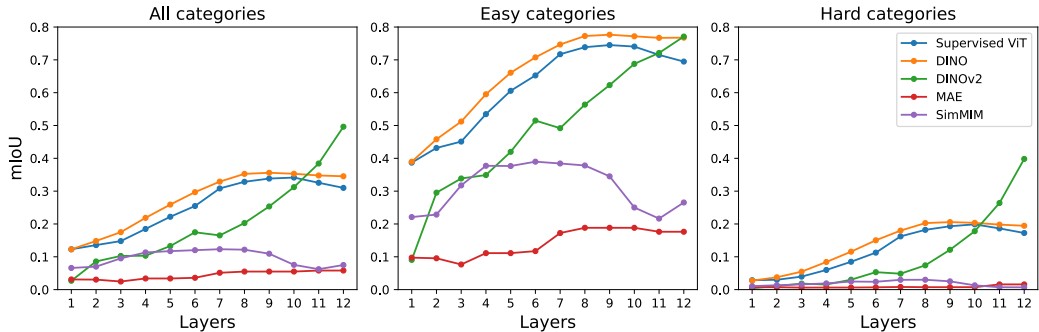

Figure 2: k-NN patch classification performance (mIoU) for various vision transformers on few-shot subset of Cityscapes.

**k-NN and linear probing.** We use two simple classifiers to analyze the underlying representations: k-NN with $k = 1$ and fitting a linear softmax classifier. Both are trained on the patch representations of the few-shot training set. The motivation for taking these two basic methods is to understand whether the patches of a given object category cluster together or are linearly separable from other object categories in the representation space. In case of MAE, we follow recommendations from He et al. (2022) (which on its turn refers to Doersch et al. (2015)) and apply batch normalization to the extracted features before the linear layer. We note that we get almost $2.5\times$ better results for linear probing with BatchNorm. We do not use BatchNorm for SimMIM, as adding it worsens the performance.

**The size of the training set matters for linear models, less for k-NN.** In Figure 1 we illustrate the performance of the k-NN and linear classifiers for different sizes of the training dataset. We first find the linear models achieve generally get better results than the k-NN, especially with more training data. However, for the DINO versions and the supervised ViT, the k-NN and linear classifiers still perform comparable to each other. Instead, the MAE representations yield surprisingly bad k-NN performance. While its linear results are only slightly inferior to its DINO counterpart, the k-NN classifier leads to a remarkable about 4 times worse performance.

**Distinct layer-wise behavior across ViTs.** Next, we perform a detailed analysis of the k-NN patch classification performance for representation extracted at different layers of the network in fig. 2. Supervised ViT and DINO perform strikingly similar. The performance slowly improves from the first layer to the 8-th layer and then saturates. There is a slight drop of performance in the last two layers, which is more visible in case of Supervised ViT.

DINOv2's behavior is quite different. In the first layers its performance is worse than DINO. For the easiest five object categories (*road*, *vegetation*, *sky*, *car*, *building*) the performance catches up in the last layer. For the other, harder object categories DINOv2 catches up with DINO and Supervised ViT in the 10th layer, and then significantly outperforms them in the 11th and 12th layers. In particular, IoU of *bus* category jumps from 0.059 on the 9th layer to 0.729 on the 12th layer. The advantage of DINOv2 thus mostly stem from the harder categories.

Again, we observe poor k-NN performance in case of the MAE. SimMIM, another ViT trained to reconstruct images, performs better than MAE, but only in the middle layers. The difference is more significant for easier object categories. The quality of the last three layers is similar to MAE.

**Evaluating robustness to degradations.** According to Oquab et al. (2023), DINOv2 is extremely robust compared to other pretrained vision transformers as measured by its performance on domain-shifted versions of ImageNet. In this subsection, we perform such an analysis, but on the image patch representation level instead.

We applied three sets of corruptions to the same small subset of the validation set of Cityscapes. First, we blur out the images by applying four sizes of the blur kernel: $10 \times 10$, $20 \times 20$, $30 \times 30$ and $40 \times 40$. Second, we add four different levels of Gaussian noise (with mean $0$ and standard deviations $10, 20, 30$, and $40$). At last, we add frequency-based random noise (with $40$ standard deviation) to the images the following way. We generate a Gaussian noise of the same shape as the source image,

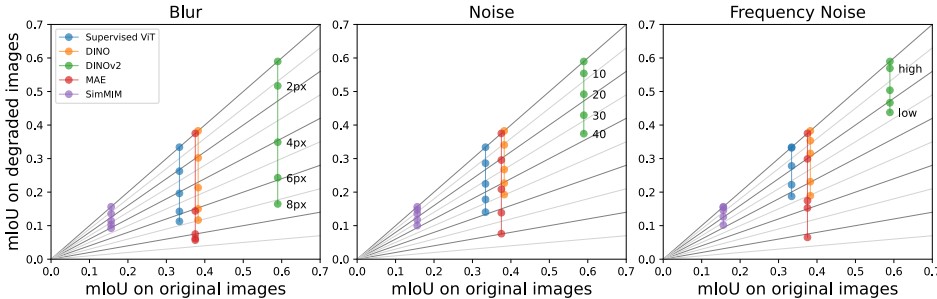

Figure 3: Robustness of linear models for segmentation on Cityscapes. For every model (differing by colors) we have five dots, each representing one level of degradation. The y-axis shows the mIoU on corrupted images. The top most dot in each vertical line corresponds to the original images. Grey diagonal lines indicate levels of equal relative drop in mIoU.

convert it into the frequency domain, keep the frequencies only inside one of the four narrow bands, convert the noisy image back to image pixels, and add it to the original image. Figure 9 in Appendix visualizes this process. We feed these degraded images to the ViTs and check the robustness of the methods for the different types and levels of degradations. The first row of fig. 3 demonstrates the results of kNN for blurred, additive Gaussian noise and added frequency-based random noise for each color, respectively.

One can claim that the robustness to various degradations may be attributed to the augmentations used during training. Specifically, models that employ color enhancement augmentations are assumed to exhibit higher resilience against those types of degradations (blur, Gaussian noise, etc). However, according to He et al. (2022) color jittering-based augmentation degrades its performance. This suggests a potential trade-off between performance and robustness to degradations. Due to computational limitations, further investigation of this hypothesis is left for future work.

**DINOv2 is more robust when tested on small degradations.** Figure 3 shows that DINOv2 is relatively more robust for the smallest blur radius compared to DINO and Supervised ViT. For stronger blurred versions these three models degrade by almost the same ratio. MAE degrades relatively faster than others. k-NN results are similar to linear probing results, except for MAE, for which even the smallest degradation leads to a collapse in predictions: k-NN predicts the same class for all patches (usually *vegetation* or *sky*).

**Supervised ViT does not suffer from high-frequency noise.** According to Park et al. (2023), masking-based methods like MAE rely more on high frequency features, while the methods based on contrastive training (including DINO) rely more on lower frequency features. This implies that DINO representations should be more robust wrt high frequency noise, while MAE representations should perform better under low frequency noise. In our experiments, MAE performs worse for all frequencies of noise. Instead, Supervised ViT is 100% robust with respect to high frequency noise. This can be explained by its objective to learn object categories of the full image which makes its last layers to forget irrelevant high-frequency information. A similar phenomenon was reported in Kleinman et al. (2021).

**The findings are confirmed on ADE20K.** We have created a similar few-shot subset of ADE20K training data which consists of 600 images in the training set (4 images per class) of size 672x448 and another 300 images in the validation set. With both k-NN and linear probing we get similar relative performance among the ViTs we have tested (table 2 in appendix A.3).

## 4 WHY RECONSTRUCTION-BASED MODELS PERFORM SO POOR IN K-NN?

Throughout our experiments we discovered that MAE's patch embeddings perform reasonably well with linear probing, but fail with k-NN. We hypothesize that the patch embeddings of MAE might have high variance in some dimensions which could drastically increase the distances among

same-category patches and therefore harm k-NN, while not affecting the performance of (possibly small-margin) linear models.

**Variances of MAE and SimMIM features are extremely diverse.** We computed the variance for each of the features for all models. In fig. 4a, we plot the variances of all 768 features in a decreasing order. We observe that the variances throughout all features of Supervised ViT (and DINO) are relatively uniform. However, for MAE and SimMIM, there are several features with very high variances and there is a long tail of close to zero variance features.

**Improving k-NN performance for MAE.** We removed $m$ features with the highest variances and measured few-shot segmentation performance using k-NN and linear probing with the shortened embeddings. As seen in fig. 4b, with just $m = 10$ features removed, MAE's k-NN performance jumped from 0.058 to 0.170, without sacrificing the performance of linear probing. The performance of k-NN kept increasing up to 0.295 at $m = 200$. Afterwards, the scores for both k-NN and linear probing started to decline. This finding implies that around a quarter of features of MAE's embeddings are not useful for patch-level image segmentation for neither linear models nor k-NNs. On the other hand, these features comprise nearly all of the variance of the embeddings. What information do these features hold?

One hypothesis is that these features are necessary for identifying specific instances of objects among the same category, or for distinguishing fine-grained categories of objects. In Section 5 we provide negative evidence for this hypothesis: removing high-variance features improves retrieval performance so much that the negative effect of losing some instance-specific information, if true, is not detectable.

**Image reconstruction quality is one metric that suffers.** Another hypothesis is that this information is useful for pixel-level reconstruction of the patches. To verify this, we took MAE embeddings with $m = 200$ features replaced with zeros, and used the pretrained decoder to reconstruct the image. Reconstruction error indeed increases with the replaced features (table 4 in Appendix). It is still unclear why the variance of those features is high.

**High variance features do not store global context.** Another hypothesis is that the high variance features might contain global information about the image not directly relevant to the semantic class of the current patch. An example of a global information is the number of pixels of each class in the entire image. To test whether patch representations contain such information, we use L2-regularized linear regression to predict those numbers (normalized by the total number of pixels) for the three most common classes (road, building, vegetation) from the representation of a central patch. We report $R^2$ for each class and average them. We see that MAE embeddings without top 200 high-variance features have the same predictive power (0.672) as the full MAE embeddings (0.676). It implies that the high variance features do not contain additional information about the global context not present in the other features. Furthermore, the features with the highest variances are less predictive (0.550) than a random subset of 200 features (0.666); see Table 6 of Appendix.

**ViTs with no reconstruction loss do not have this phenomenon.** We performed similar analyses for all other ViTs, and visualize the results on fig. 4c. The phenomenon of improved k-NN performance when high-variance features are removed is present only in the models trained with pixel-level reconstruction objective. In case of other ViTs, while removing high-variance features does not improve k-NN performance, it does not harm it either. Linear probing performance also stays robust with these removals up to some level.

**Feature normalization has a similar effect for MAE, but not for SimMIM.** We created another version of MAE embeddings by applying the pretrained BatchNorm layer from the linear model, before passing them to k-NN classifier. It helped to improve k-NN performance by almost the same amount as removing $m = 200$ features. It improved SimMIM's k-NN performance as well, but linear probing worsened significantly. We conclude that feature normalization is an alternative but not an identical strategy to feature removal for minimizing the negative impact of high-variance features.

**High variance features of MAE are relatively stable across datasets.** We identify top 200 high-variance features of patch representations extracted from Cityscapes, ADE20K and FAIR1M datasets. 196 of these features are shared among Cityscapes and ADE20K, while 192 of them are shared among Cityscapes and FAIR1M.

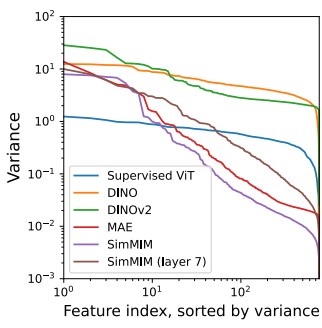 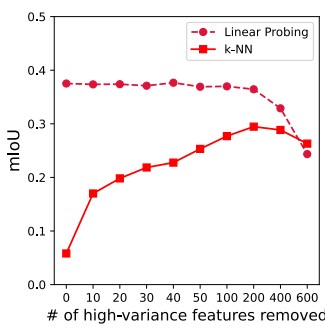 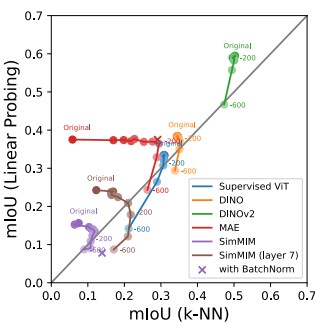

(a) Variances of each of the 768 dimensions of embeddings of various ViTs sorted in decreasing order. Note that both axis are logarithmic.

(b) k-NN and linear probing performance of MAE embeddings with $m$ highest-variance dimensions removed.

(c) k-NN and linear probing performance of embeddings of various ViTs with $m$ highest-variance dimensions removed.

Figure 4: Analysis of high-variance features of ViT patch embeddings.

## 5 AT WHICH GRANULARITY TRANSFORMERS DETECT OBJECTS?

In section 3 we analyzed the performance of ViTs for the object-level few-shot semantic segmentation task. This involves assigning a class to each patch that contains predefined objects, such as airplanes and cars. This led to the next question of whether ViTs can learn to distinguish between objects of the same category but of different types (such as types of cars or types of airplanes) or whether ViTs can perform well in distinguishing the same exact object when other instances of it are available.

**Methodology.** We take FAIR1M Sun et al. (2022), a large dataset of satellite imagery designed for fine-grained object detection. Note that none of the ViTs we tested were pretrained on satellite imagery (even DINOv2, as far as we can tell). The objects in FAIR1M are annotated with respect to 5 supercategories: airplane, ship, vehicle, court, and road and 37 fine-grained categories (types of airplanes, types of ships, etc.). The annotations are in the form of rotated bounding boxes (without pixel precision). However, to the best of our knowledge, all object instances appear on only one image.

The images have varying sizes, usually larger than $1000 \times 1000$px. We cropped all of them to $224 \times 224$px tiles and kept only 8 images per each fine-grained category. This guarantees that our dataset contains at least 8 instances of each category, but in fact it contains many more instances of the common objects. Table 3 in Appendix lists dataset statistics. We have 295 images with 196 patches each for ViT-B/16 models, and 256 patches each for DINOv2.

Next, we created transformed versions of all images by applying diagonal shifts by $1, 2, 3$, and $4$ pixels. We also applied the blur and Gaussian noise degradations. We computed the patch representations of all these images. This creates many images with the same instances of objects.

For each patch of a transformed image covering an annotated object, we retrieve the closest patch from the complete set of original image patches. Ideally, the closest patch should be the original version of the patch (without blur, noise, or shift). Otherwise, the second best option should be another instance of the same fine-grained category, the third best option would be a patch of an object within the same supercategory. The worst-case scenario occurs when the closest patch belongs to another category or is a background patch. For every model and image transformation level we calculate the number of patches for which the closest patch belongs to the mentioned classes.

**Increased levels of image degradation decrease all metrics.** The results are presented in Figure 5. When the target patch transformation is small (e.g. Gaussian noise with just 10 pixels standard deviation), the closest patch is almost always the original one for all models. One notable exception is MAE, for which in case of roughly 40-50% patches the closest one is the right one, but for the remaining ones the closest patch is a background patch. With stronger transformations, the ratio of correct patches decreases for all models, and more than half of the remaining patches are matched to patches of the same fine-grained object category (again, with the exception of MAE).

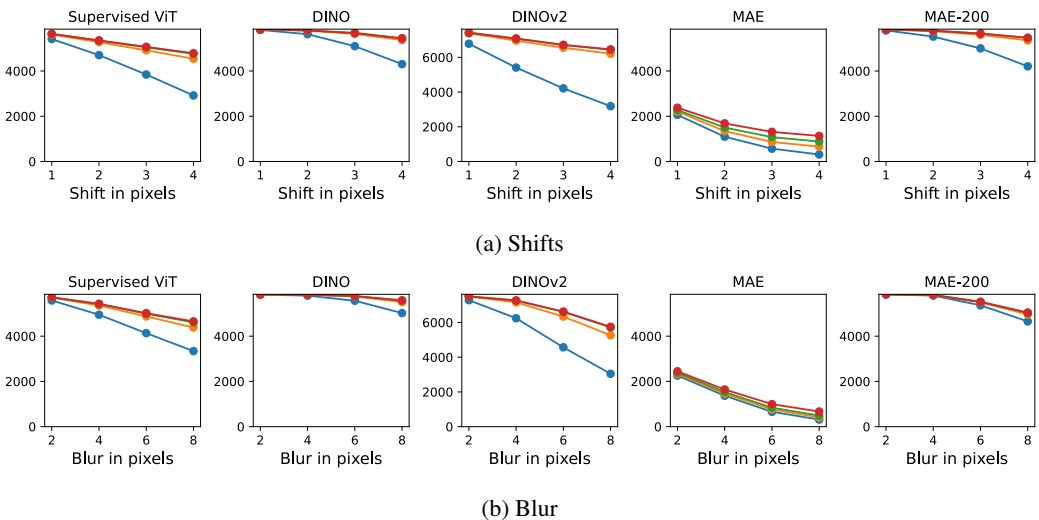

(a) Shifts

(b) Blur

Figure 5: Stacked area plots for each model (columns) and transformation type (rows). In each subplot, x-axis is the level of transformation, y-axis is the percentage of correctly matched patches in each granularity.

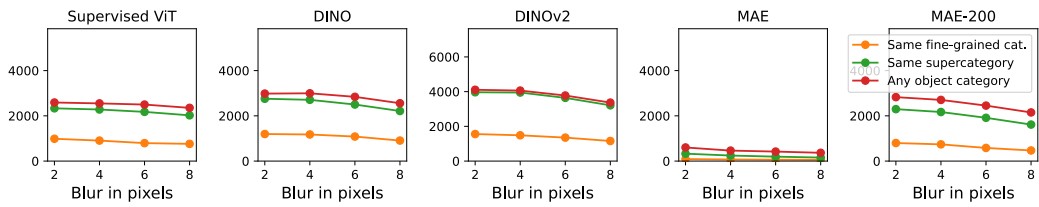

Figure 6: Patch retrieval results when the original image is not available. In each subplot, x-axis is the level of blur, y-axis is the number of correctly matched patches in each granularity.

**DINO is the most robust one.** Among the ViTs we tested, DINO is the most robust one across all transformations. Surprisingly, DINOv2 is less robust by all metrics, and performs similar to Supervised ViT. To verify whether this disadvantage of DINOv2 comes from the patch-level loss term or the scale of the model and the dataset, we perform the same analysis with iBOT embeddings. iBOT performs even slightly better than DINO (fig. 10 in Appendix), which means that patch-level loss cannot be blamed.

**Retrieving patches from other images.** We notice that most of the retrieved patches come from the same image tile. One possible explanation is that the patch embeddings contain image-level information. We repeat this experiment with the patches of the original image removed from the set of available patches. In this setting the closest patch can be of the same fine-grained category, same supercategory, incorrect supercategory and can be a background patch. Here DINOv2 takes the lead, Supervised ViT and DINO perform slightly worse, and MAE performs poorly (fig. 6).

**MAE's performance can be improved**. We repeated this experiment with the shortened embeddings of MAE (200 highest-variance features removed). This version of MAE performs significantly better and outperforms both DINOv2 and Supervised ViT, and falls short only of DINO. This result implies that removing the high-variance features of MAE embeddings not only helps in semantic segmentation, but also in the identification of specific object instances in transformed images. On the other hand, this adds to evidence that the high-variance features do not hold any unique information necessary for identifying the instances, as seen by almost ideal performance on the patches with small transformations (fig. 5).

## 6 How Well ViTs Track Objects?

Object tracking requires identification of the same object instance across frames in a video. In this section we analyze the robustness of patch embeddings over time, as the objects undergo appearance changes.

**Methodology.** We take track-validation set of MOT 2020 images subset of BDD-100K dataset Yu et al. (2020) and extract patch representations from each frame. We use ROIAlign He et al. (2017) to pool features of each object instance based on its ground truth bounding box. For a given time difference $\Delta$ between two frames and for each frame at time step $t$, we retrieve the object with the closest embedding vector from the frame at time step $t + \Delta$. We report the ratio of correctly retrieved instances as the main metric. We also report the percentage of object instances for which the retrieved instance has the same object category. Results are shown in fig. 7.

**DINO-like models are best at tracking.** The accuracy of all models degrades for longer time intervals, as appearance changes in average increases over time. The retrieval performance stays relatively constant for intervals larger than one second, i.e. about 30 frames. DINO and DINOv2 show the best performance, with a slight advantage for DINO for $\Delta < 10$, which is consistent with the patch retrieval experiments in section 5. Surprisingly, the Supervised ViT is substantially inferior to the DINO, and instead performs similar to MAE.

Figure 7: Object instance matching performance for various ViTs on the BDD100k MOT dataset. We report the ratio of correctly retrieved instances as a function of the frame gap $\Delta$. Solid lines with circles represent accuracy for instance retrieval, while dashed lines with triangles show the accuracy for retrieval within the same category.

**Incorrect matches usually have the correct category.** When instance matching fails, the closest object belongs to the same object category in more than 95% of cases for DINOv2, slightly less for DINO and Supervised ViT. MAE's performance is noticeably worse than Supervised ViT. These results are consistent with patch classification metrics from section 3.

We repeat the experiments on MOT17 dataset and see a similar behavior. Contrastive approaches strongly outperform ViTs based on masked image modeling. The results are in appendix A.3.

## 7 Conclusion and Limitations

We perform a comprehensive analysis and comparison of the quality and properties of locality patch embeddings extracted from self-supervised ViT models. We observe that the contrastive learning based DINO series outperform both supervised and masked image modeling approaches. Moreover, we identify and study the inferior k-NN classification performance of MAE which limits its use without fine-tuning. We find that the features with relatively high variances are not informative for patch classification or retrieval tasks and that their removal results in improvement for k-NN performance while not harming linear probing.

**Limitations.** Due to the high computational cost, we are unable to retrain networks, preventing us from analyzing architectural choices or loss components in the discussed Vision Transformers (ViTs). Therefore, our comparison is restricted to existing pre-trained networks.

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
