## A APPENDIX

### A.1 THE CHOICE OF SELF-SUPERVISED VISION TRANSFORMERS

We use the following pretrained visual transformers in our analysis.

**DINO** (Caron et al., 2021) is a self-supervised ViT that utilizes a self-distillation (student-teacher) framework. Different augmented versions of the same image pass through the teacher and student networks, and the student network is optimized to produce the same [CLS] vector as the teacher. Teacher's weights are then updated from the student's weight using exponential moving average.

**Masked Autoencoder (MAE)** (He et al., 2022) is trained to reconstruct the original image given partial observations. During training, a large random fraction of image patches are masked out in the input. The encoder is applied to the visible patches only. A relatively lightweight decoder gets encoder's outputs as input, along with [MASK] tokens for the masked patches, and attempts to reconstruct the original image. We use the pretrained encoder as a feature extractor for patches.

**SimMIM** (Xie et al., 2022) is another framework for vision transformers that uses masked image modeling. The main difference from MAE is that SimMIM uses a simple linear decoder on top of encoder's outputs.

In one experiment we analyzed **iBOT** (Zhou et al., 2021), which is another teacher-student framework that additionally masks some of the patches for the student network. In addition to DINO's objective, it has another loss term that forces the student network to produce patch representations for the masked patches similar to ones given by the teacher on an unmasked image.

**DINOv2** (Oquab et al., 2023) is a recent extension of iBOT which was trained on a much larger dataset. The dataset is comprised of 17 million images from ImageNet-2, Mapillary SLS and Google Landmarks v2, and additional 125 million images retrieved from a large pool of web-crawled images with the condition of being similar to images to a pre-selected 27 publicly available datasets. The main model has more than 1B parameters, which forced the authors to use multiple regularization techniques to stabilize the training. They also provide the distilled versions of the main model, which we use in our work.

Finally, we used a supervised baseline trained on ImageNet-1k with image-level labels. A linear layer was trained on top of the [CLS] token. Throughout the paper this one will be called **Supervised ViT** (Dosovitskiy et al., 2021). [1]

All methods have been applied to multiple sizes of ViTs. In this work we focus only on one size that is available for all methods: ViT-B/16 with 86 million parameters. DINOv2 is the only one that does not have a ViT-B/16 version. Instead, we used the closest one: ViT-B/14, which is distilled from the ViT-g/14 model. This is another distinction between DINOv2 and others: the patches are a little bit smaller, and an image of size 224x224px has a larger number of DINOv2 patches.

The models also differ by the types of data augmentation used during pretraining. MAE used only simple resized crops and flips. DINO additionally used color jitter and blur with some differences between teacher and student networks. Supervised ViT uses a bunch of tricks as part of RandAugment (Cubuk et al., 2020), and also uses Mixup (Zhang et al., 2018). DINOv2's augmentations are similar to DINO. More details are available in table 1.

We pass images through these ViTs and extract all patch embeddings from the 12-th layer. All ViTs apply layer normalization (Ba et al., 2016) on top of these embeddings. For consistency, we also apply layer normalization when we extract embeddings from the inner layers of ViTs.

### A.2 DATA AUGMENTATIONS USED IN VITS

All ViTs we tested used data augmentation during the pre-training phase. In this section, we discuss the differences of augmentation strategies used.

In **DINO** Caron et al. (2021) and **DINOv2** Oquab et al. (2023) an image is cropped to two global crops or views for teacher network and multiple local views for student network. They apply different

---

[1]We used a version from *torchvision* package: `https://pytorch.org/vision/0.15/models/generated/torchvision.models.vit_b_16.html`.

augmentations for different views. **MAE** He et al. (2022) applies cropping-only augmentations. See the Table 1 for more details. For resized crop all models choose 224 for output size.

**Supervised ViT** Dosovitskiy et al. (2021) combines following techniques for data augmentation following to Steiner et al. (2021)

- Mixup Zhang et al. (2018) with $\alpha = 0.2$, where $\alpha = 0$ means no Mixup.
- TensorFlow impementation of RandAugment Cubuk et al. (2020) with magnitude parameter $m = 15$ and number of augmentation layers $l = 2$.

In table 1 we summarize the augmentation details.

| | DINO | DINOv2 | MAE |
|---|---|---|---|
| **Resized Crop** | *For global views* $scale = (0.4, 1.0)$ *For local views* $scale = (0.05, 0.4)$ | *For global views* $scale = (0.32, 1.0)$ *For local views* $scale = (0.05, 0.4)$ | $scale = (0.2, 1.0)$ |
| **Gaussian Blur** | *For first global view* $p = 1.0$ *For second global view* $p = 0.1$ *For local views* $p = 0.5$ | *For first global view* $p = 1.0$ *For second global view* $p = 0.1$ *For local views* $p = 0.5$ | $-$ |
| **Solarization** | *For second global view* $p = 0.2$ | *For second global view* $p = 0.2$ | $-$ |
| **Horizontal Flip** | $p = 0.5$ | $p = 0.5$ | $p = 0.5$ |
| **Gray Scale** | $p = 0.2$ | $p = 0.2$ | $-$ |
| **Color Jittering** | $p = 0.8$ | $p = 0.8$ | $-$ |

Table 1: Data augmentations for pretrained models

### A.3 RESULTS ON MORE DATASETS

We have additionally conducted patch classification experiments on ADE20K. As seen in table 2, the ranking of the various ViTs are similar in both k-NN and linear probing settings.

Figure 8 shows the results of the tracking experiment performed on MOT17 dataset.

### A.4 STATISTICS OF THE FEW-SHOT VERSION OF FAIR1M DATASET

We created a subset of the FAIR1M training set in a way that ensures each fine-grained object category appears in at least eight images. We cropped the original images to $224 \times 224$ px tiles and for each

| | CityScapes | | ADE20K | |
|---|---|---|---|---|
| | 1-NN | Linear | 1-NN | Linear |
| **Supervised ViT** | 0.310 | 0.33 | 0.053 | 0.067 |
| **DINO** | 0.345 | 0.383 | 0.055 | 0.070 |
| **DINOv2** | 0.496 | 0.590 | 0.136 | 0.161 |
| **MAE** | 0.058 | 0.375 | 0.006 | 0.066 |
| **MAE200** | 0.295 | - | 0.043 | 0.056 |
| **SiMIM** | 0.075 | 0.156 | 0.006 | 0.032 |

Table 2: Patch classification results on CityScapes and ADE20K datasets.

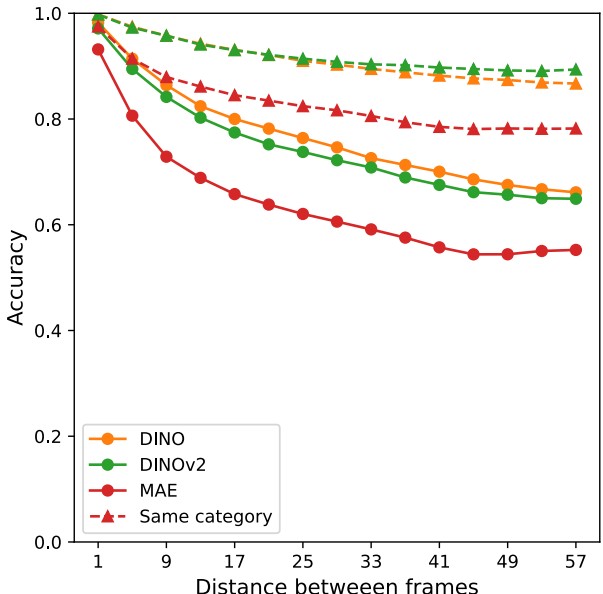

Figure 8: Object instance matching performance for various ViTs on the MOT17 dataset.

tile we kept the list of object categories that are present in the tile. We consider category A to be present in a tile if at least one rotated bounded box of type A has at least 1/3 of its area inside the tile. For each fine-grained category, we took eight images that contain an object of that category. Then we remove those images from the cohort and proceed to the next object category. This way we collected $37 \times 8 - 1$ images, because there were only 7 tiles for one particular fine-grained category (*bus*). Table 3 shows the number of patches of each category in our few-shot set of 295 images. Note that DINOv2 has 256 patches per image, while all others have 196 patches per image.

## A.5 RECONSTRUCTION ERROR ANALYSIS FOR MAE

To understand what information is stored in the high variance features of MAE, if removing them does not harm patch classification or patch retrieval performance, we conduct experiments with image reconstruction. The hypothesis suggests that the removed features play a role in certain reconstruction properties. We use the pretrained decoder of MAE in two settings: when no patches are masked and when 75% of the patches are masked. In table 4, one can see that when the high variance features are filled with zeros, the reconstruction metrics get slightly worse. This indicates that these features contain knowledge on how to reconstruct the image, but they are not essential for most other downstream tasks. The accuracy of the reconstruction is evaluated using Mean Square Error (MSE), Peak Signal-to-Noise Ratio (PSNR) and Structural Similarity Index (SSIM) Wang et al. (2004) metrics.

## A.6 FREQUENCY NOISE COMPUTATION

In fig. 3 we presented some degradation analyses and results for various degradations, including frequency-based random noise. In this section, we briefly expand on the experiments and go over their setup.

To create frequency-based random noise, we first generated 2D Gaussian random noise with the same dimensions as the image across all three color channels. We then applied a Fourier transform to the noise, masked it in the frequency space, and applied an inverse Fourier transform to obtain the frequency-based random noise. The following formula demonstrates the addition of frequency-based random noise to the image:

| Supercategory | Fine-grained category | Pixels | Patches | Patches DINOv2 |
|---|---|---|---|---|
| | background | 13305235 | 51970 | 67897 |
| airplane | boeing737 | 36151 | 139 | 183 |
| | boeing777 | 57034 | 228 | 290 |
| | boeing747 | 56100 | 218 | 289 |
| | boeing787 | 46713 | 178 | 234 |
| | a320 | 0 | 0 | 0 |
| | a220 | 42522 | 159 | 220 |
| | 576 | 787 | 269 | 350 |
| | a350 | 100574 | 398 | 506 |
| | a321 | 57768 | 228 | 293 |
| | c919 | 14654 | 57 | 74 |
| | arj21 | 11891 | 51 | 67 |
| | other-airplane | 62174 | 246 | 312 |
| | total | 554797 | 2171 | 2818 |
| ship | passenger ship | 26988 | 101 | 134 |
| | motorboat | 28031 | 118 | 142 |
| | fishing boat | 30738 | 127 | 160 |
| | tugboat | 10226 | 42 | 55 |
| | engineering ship | 88900 | 342 | 443 |
| | liquid cargo ship | 24148 | 91 | 122 |
| | dry cargo ship | 144103 | 562 | 732 |
| | warship | 28255 | 114 | 142 |
| | other-ship | 15820 | 63 | 81 |
| | total | 397209 | 1560 | 2011 |
| car | small car | 24371 | 98 | 125 |
| | bus | 2451 | 8 | 13 |
| | cargo truck | 22516 | 92 | 119 |
| | dump truck | 8553 | 37 | 44 |
| | van | 33960 | 131 | 165 |
| | trailer | 7654 | 32 | 39 |
| | tractor | 2734 | 10 | 16 |
| | truck tractor | 3760 | 17 | 17 |
| | excavator | 10048 | 38 | 52 |
| | other-vehicle | 8138 | 37 | 47 |
| | total | 124185 | 500 | 637 |
| court | baseball field | 108367 | 421 | 556 |
| | basketball court | 16005 | 72 | 82 |
| | football field | 87005 | 338 | 438 |
| | tennis court | 57114 | 212 | 294 |
| | total | 268491 | 1043 | 1370 |
| road | roundabout | 94068 | 360 | 490 |
| | intersection | 38180 | 144 | 190 |
| | bridge | 19755 | 72 | 107 |
| | total | 152003 | 576 | 787 |

Table 3: Patch statistics of the few-shot FAIR1M subset we used in our experiments.

|  |  | Original Reconstruction | Without high variance features | Without low variance features |
|---|---|---|---|---|
| MSN ↓ | No masking | 0.163 | 0.199 | 1.173 |
|  | Mask ratio = 75% | 0.090 | 0.103 | 1.164 |
| PSNR ↑ | No masking | 25.666 | 23.349 | 15.350 |
|  | Mask ratio = 75% | 25.571 | 24.488 | 14.474 |
| SSIM ↑ | No masking | 0.78 | 0.722 | -0.074 |
|  | Mask ratio = 75% | 0.786 | 0.737 | -0.064 |

Table 4: Reconstruction metrics for MAE with and without replacement of 200 highest variance and 200 lowest variance features with zeros.

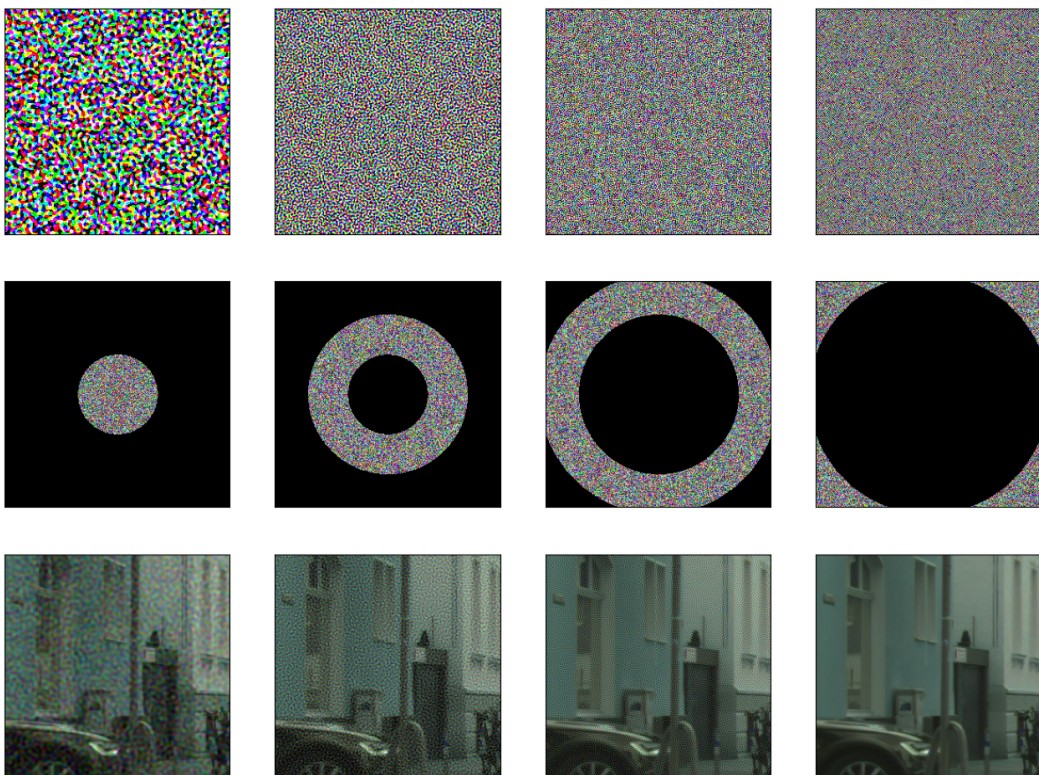

Figure 9: The first row shows the additive noise, the second row corresponds to the masked noise in the frequency domain, and the last row corresponds to the image with additive frequency-based noise.

$$I = I_0 + \mathcal{F}^{-1}(\mathcal{F}(\epsilon) \odot \mathbf{M}_f),$$

where $I_0$ corresponds to the original image, $\epsilon$ corresponds to random noise with the same dimensions as the image. Each pixel of the noise follows a Gaussian distribution with a mean of $0$ and a given variance, $M$ represents the frequency mask shown in fig. 9, we also shifted the zero-frequency component to the center of the spectrum and then applied an inverse shift. At last, $\mathcal{F}$ and $\mathcal{F}^{-1}$ correspond to the Fourier transform and inverse Fourier transform, respectively. The additive frequency-based random noise, the corresponding mask, and the noisy images for four different makes are demonstrated in fig. 9 for more details.

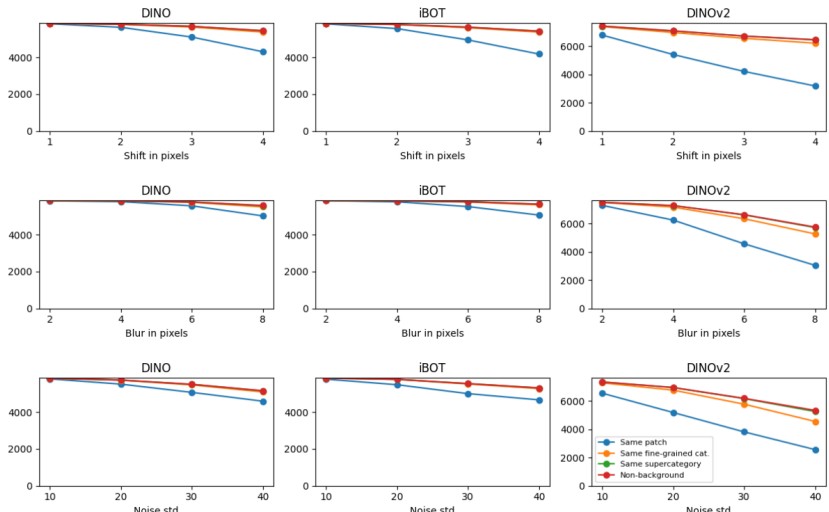

Figure 10: iBOT's retrieval performance is slightly better than DINO, and is much better than DINOv2.

## A.7 ADDITIONAL EXPERIMENTS ON FAIR1M

### A.7.1 DINO VS. IBOT VS. DINOV2

The most unexpected result of section 5 is that DINO representations are better for retrieving the closest patch given a corrupted patch than DINOv2 representations. DINOv2 has a series of differences compared to DINO. These differences can be split into two categories: those related to the loss terms and those related to the scale of the model and the dataset. The new patch-level loss term of DINOv2 first appeared in iBOT. Here we performed the same set of experiments on iBOT as well, to compare it with DINO and DINOv2. As seen in fig. 10, iBOT performs at least as good as DINO. This means that the new loss term cannot be blamed for the worse retrieval performance for DINOv2.

### A.7.2 MORE EXPERIMENTS FOR FAIR1M

We explore one more type of image transformation suitable for satellite images: rotation. We rotate each $224 \times 224$ tile by 5, 10, 15 and 20 degrees counter clockwise and pass them through the ViTs to obtain representations of the rotated patches. Then for each patch we look for the closest patch of the original (non-rotated) images. Here the first level of evaluation, called "same-patch" is not straightforward to define, as it is not obvious which is the corresponding patch in the original image. We define the corresponding patch as the one which contains the center point of the rotated patch. Obviously, several patches near the corners of the rotated tile will not have a corresponding patch. This puts an upper bound on the same-patch retrieval accuracy. fig. 11 shows the results along with the upper bounds. It is obvious that all models are significantly less robust with respect to even 5 degree rotation than with the highest blur radius or noise level we tried. The order of models in terms of performance is similar to other image transformations: DINO is the best one, followed by MAE with 200 high-variance features removed, followed by DINOv2 and Supervised ViT, and MAE is a distant outlier.

## A.8 DISCUSSION ON TILING

Since most ViTs are trained on small-sized images and their original weights are provided for $224 \times 224$ images without interpolating positional embeddings, preprocessing becomes necessary when working with larger images. For example, in the case of the FAIR1M dataset, the images are $1000 \times 1000$, and for CityScapes the images are $1024 \times 2018$. There are several options for handling such images, including rescaling them to a smaller size, tiling them (dividing them into smaller pieces and conducting experiments on these tiles, then combining them to reconstruct the original image size), or forcing the ViT to process the full-sized image by interpolating the positional embeddings.

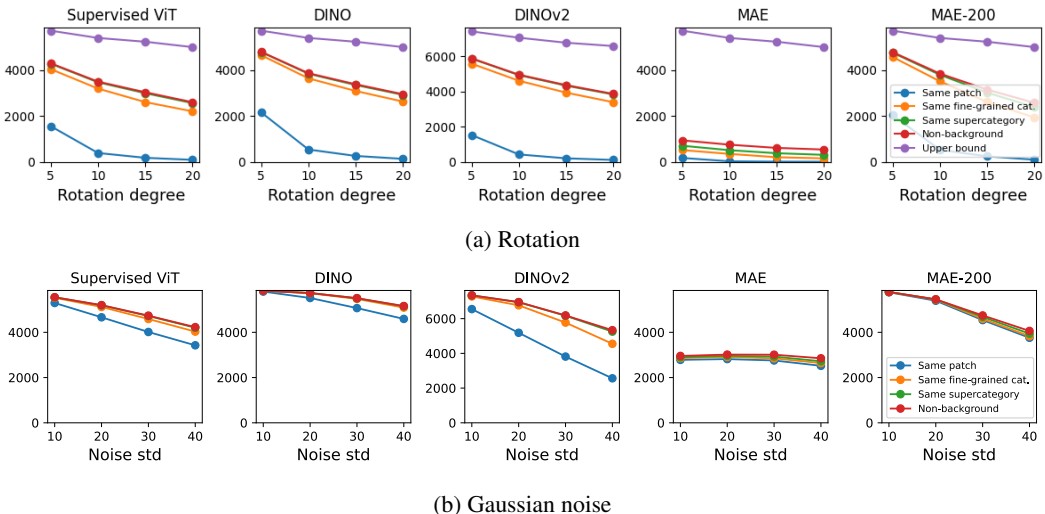

(a) Rotation

(b) Gaussian noise

Figure 11: Stacked area plots for each model (columns) on rotated images and on images with Gaussian noise. In each subplot, x-axis is the level of transformation, y-axis is the percentage of correctly matched patches in each granularity.

|        | Input Size        | Linear    | KNN       |
|--------|-------------------|-----------|-----------|
| **DINO** | $256 \times 256$   | **0.383** | **0.345** |
| **DINO** | $1024 \times 2048$ | 0.340     | 0.308     |
| **MAE**  | $256 \times 256$   | **0.375** | **0.058** |
| **MAE**  | $1024 \times 2048$ | 0.176     | 0.02      |

Table 5: Patch classification performance on Cityscapes depending on tiling of the input images.

The risk of tiling the images is that smaller tiles might lose the global context of the image, which is an important factor in transformer architecture. In all experiments in the paper we chose the tiling method. Here we explore the effect of using full-scale images.

### A.8.1 CITYSCAPES

We conduct the following two experiments. For the first experiment, we tile the images into $256 \times 256$ patches, resize them into $224 \times 224$, and separately compute the embeddings for corresponding ViTs (only for DINO and MAE). For the second experiment, we compute the patch embeddings for the full images. We check the accuracy of semantic segmentation for CityScapes dataset on a validation set of 30 images. Surprisingly, both MAE and DINO for linear probing and k-NN achieve higher mIoU values in the tiled setting. The results are summarized in table 5. In conclusion, despite the expectation that the global context of the image would contain more information, the degradation of the performance due to large input sizes is too strong.

|            | MAE    | MAE200 | MAE Random | MAE Junk |
|------------|--------|--------|------------|----------|
| road       | 0.606  | 0.603  | 0.588      | 0.551    |
| building   | 0.698  | 0.711  | 0.678      | 0.509    |
| vegetation | 0.723  | 0.700  | 0.687      | 0.591    |
| mean       | 0.6754 | 0.671  | 0.651      | 0.550    |

Table 6: $R^2$ for MAE for different classes for global context understanding.

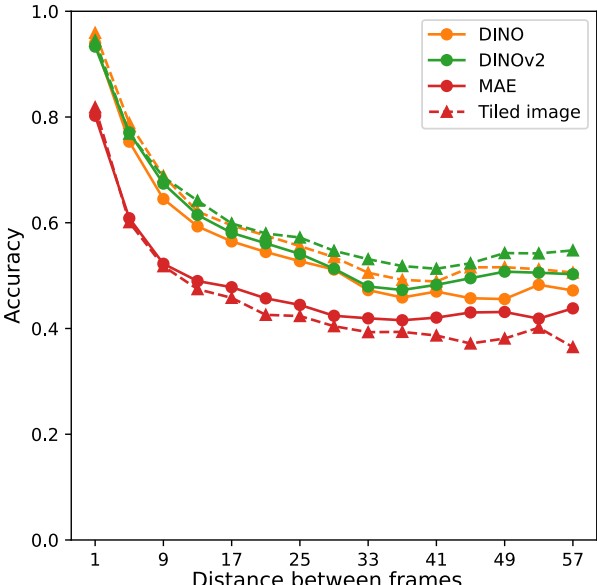

Figure 12: Comparison of full image and tiled versions of MAE and DINO in the object tracking experiment

### A.8.2 TRACKING

We perform a similar experiment for the object tracking setup. Note that in this setting we pool the representations of all patches inside a bounding box. If the object is split into multiple tiles of the same image, the averaging will occur over patch embeddings from different tiles. In fig. 12 we see the above phenomenon for DINO and DINOv2, tiled images perform better. For MAE we see a surprising result, object representations pulled from embeddings of the full image perform better.

Note that all experimental results in the object tracking experiments in this paper are reported on a subset of 4 videos from BDD-100k.

### A.9 SAMPLE PREDICTIONS ON CITYSCAPES

In fig. 13, we present qualitative results of how semantic segmentation looks based on the ViTs used in this work. We fixed two images from the Cityscapes dataset. See the first and third figures of the first row in fig. 13 for the examples and their corresponding original masks, which can be seen in the second and fourth figures of the first row, respectively. The second, third, and subsequent rows of fig. 13 demonstrate the segmentation masks obtained by the corresponding ViT, where the first and third columns correspond to the k-NN-based prediction and the second and fourth columns correspond to the linear probing-based prediction for the corresponding instances from the Cityscapes dataset. In these figures, we can qualitatively reconfirm our observations that MAE almost completely fails to segment the patches correctly with k-NN. However, its performance is competitive with linear probing. We also observe that MAE-200, which corresponds to the embeddings obtained by MAE without the top 200 features of the largest variance, outperforms MAE for k-NN and is almost identical to that of MAE for linear probing. As expected, DINO and DINOv2 qualitatively outperform all the other methods.

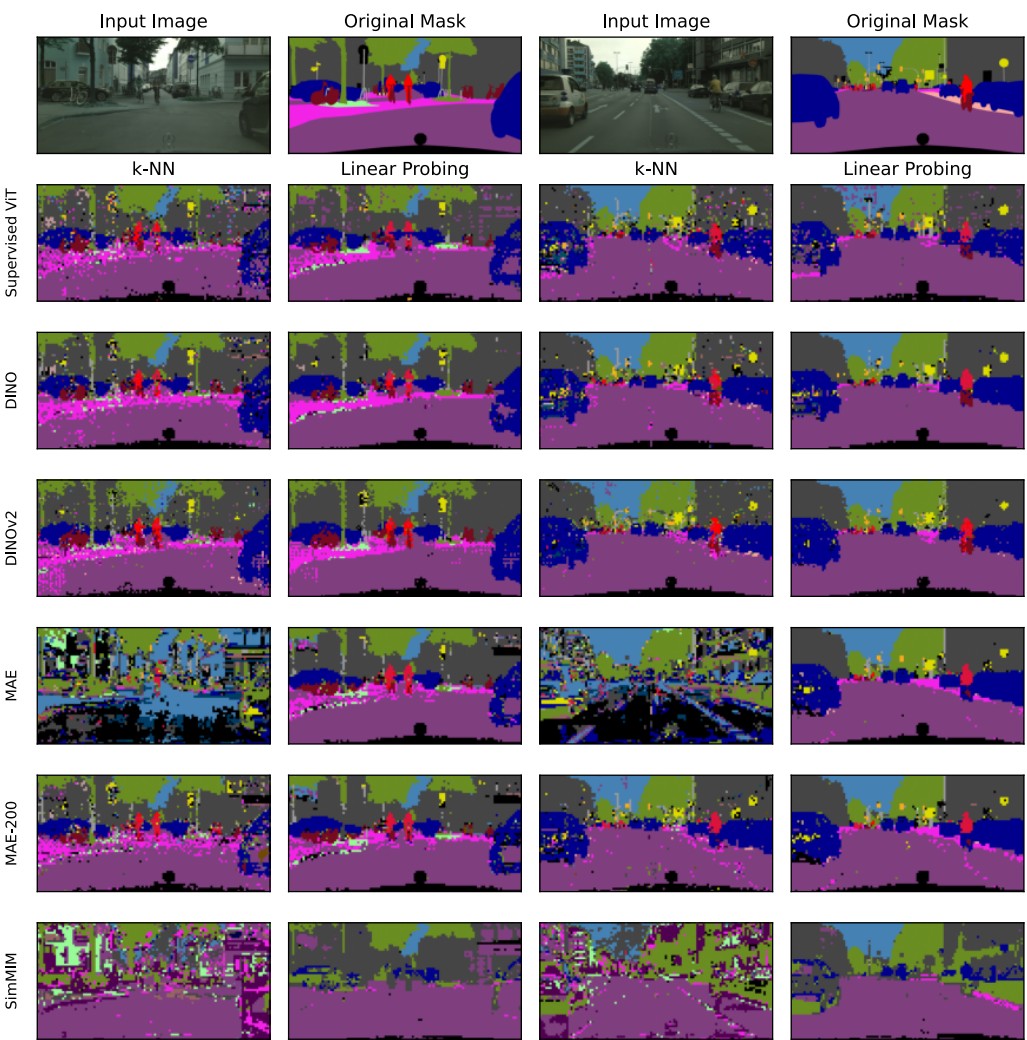

Figure 13: Predictions of k-NN and linear probing for all models for two selected images from Cityscapes.