# OpenReview forum: "Analyzing Local Representations of Self-supervised Vision Transformers"
_ICLR.cc/2024/Conference — ICLR 2024 Conference Withdrawn Submission_

### Official Review · Reviewer_aCKC · 2023-10-29

**Soundness:** 4 excellent
**Presentation:** 4 excellent
**Contribution:** 4 excellent
**Rating:** 6
**Confidence:** 4

**Summary:**

The paper centers its attention on empirically analyzing the local representation capabilities of vision transformers, encompassing various pre-training methods. This includes both contrastive-based pretraining methods like DINO and pretraining methods based on masked image modeling such as MAE. The evaluation spans across a range of tasks, including segmentation, instance identification, object retrieval, and tracking. The results of these evaluations consistently point to the superiority of contrastive learning in enhancing local representation.

**Strengths:**

1. The paper offers an extensive analysis of ViT's local representation across a wide spectrum of tasks.
2. The paper delves deeply into the factors contributing to the subpar performance of pre-training methods based on masked image modeling. This analysis is substantiated by the removal of high-diversity features, further strengthening the findings.
3. The paper is well written and easy to understand.

**Weaknesses:**

1. While the paper conducts a comprehensive analysis using MAE and SimMIM as the masked image modeling methods, it's important to acknowledge that both of these methods employ pixel information as the reconstruction target. The characteristics and outcomes of using VAE tokens or CLIP features as targets remain unexplored.

2. The evaluation method employed to assess the semantic recognition ability seems to favor contrastive learning-based approaches, potentially introducing bias. For instance, the process of splitting images into separate tiles and extracting features from each tile, treating them as individual images, bears resemblance to feature extraction from augmented images in the contrastive training pipeline.

3. It's widely recognized that MAE-pretrained vision transformers exhibit subpar performance in tasks regarding knn classification, often performing worse than contrastive learning-based methods, aligning with the paper's conclusions. However, it's worth noting that with supervised fine-tuning, masked image modeling-based methods can significantly outperform contrastive pretraining-based methods. Given that pretraining-finetuning is a popular framework in current vision research, there's a need for a comparison between these fine-tuned methods and both contrastive and masked image modeling-based methods to provide a more comprehensive understanding of their relative performance.

**Questions:**

My main concerns are the first two questions in the weakness section.

---

### Official Review · Reviewer_hpmg · 2023-10-30

**Soundness:** 3 good
**Presentation:** 2 fair
**Contribution:** 1 poor
**Rating:** 3
**Confidence:** 4

**Summary:**

This work experimented several "local" tasks (patch classification, retrieval, and object association) to evaluate representations of self-supervised vision transformers. The evaluation is done with frozen features, either with kNN or with linear probing. Some observations are drown from the studies, e.g., kNNs are not so compatible with methods like MAE (with some detailed investigations), DINO produces more universal patch representations etc. Overall the paper is more like a technical report that compiles the results of several evaluations, and it is not clear what story the paper follows.

**Strengths:**

+ The paper is investigating the representations learned from several self-supervised methods. The study of such pre-trained models is needed and an important topic for computer vision.
+ I would trust the paper's results, with good illustrations presenting the results, and it is quite clear which approaches are better under which settings.
+ Since the paper is more like a technical report, I must say the experimental "comprehensiveness" is higher than an average ICLR submission. So would appreciate the authors' effort on sharing the results.

**Weaknesses:**

- My biggest concern about the paper, or rather a technical report, is the lack of a single, clear, coherent story. This makes the paper reads more like a compilation of explorations along the direction of "frozen evaluation of self-supervised representations on some local vision tasks", and less like a paper that makes a core contribution -- e.g., an observation; an approach; a dataset benchmark. There is no clear core in the work.
- Another big concern is what's the take-away from the paper, or whether such take-aways are solid/comprehensive. For example, saying that masked modeling methods are generally inferior and less universal requires *a lot more* justifications. One may ask how about the task of "imprinting" (which masked modeling methods are pre-trained to do)? Also, as the paper mentioned, DINO v2 is pre-trained on a lot more data, this makes the comparison to others less "apple-to-apple", so it is less clear whether the DINO v2 related results are a consequence of using different/more data, or a consequence of different pre-training methods. I think the paper should at least make it clear that the comparison is just on the released checkpoints, and not generally about the approaches themselves in the beginning, so it does not mislead readers.
- A lot of the claims in the paper, are rather strong (or debatable) and not sure people would 100% agree with it. For examples:
1) k-NN with no tuned parameter -- I am sure k-NN has a lot of parameters/hyper-parameters to tune, e.g., the number k which is usually set to >1 to be more robust.
2) contrastive methods with DINO -- I am not sure DINO is characterized as contrastive or not.

**Questions:**

* For kNN, would the take-aways change if the k is changed from 1 to say 100, when the setting allows? I feel k = 1 is such a restrictive setting and does not give the comprehensive picture as it claims.

---

### Official Review · Reviewer_NRBU · 2023-10-30

**Soundness:** 2 fair
**Presentation:** 4 excellent
**Contribution:** 2 fair
**Rating:** 3
**Confidence:** 4

**Summary:**

This paper presents an analysis of local patch representations generated from pre-trained Vision Transformers (ViTs). These ViTs fall into three categories: supervised (trained using ImageNet classification), self-supervised through contrastive learning (DINO), and self-supervised through reconstruction (MAE). The study involves a series of extensive experiments aimed at addressing four key questions:

Can transformers effectively classify local patches?
Why do reconstruction-based methods show suboptimal performance with K-NN?
At which granularity transformers detect objects?
How well ViTs track objects?

The findings include that DINO consistently outperforms other approaches across most of the tasks. Furthermore, it is noted that reconstruction-based methods exhibit certain high-variance features that may be detrimental to these tasks.

**Strengths:**

The paper is well-written. The experimental settings are well-detailed and the results are clearly presented.

The authors conduct numerous experiments and I appreciate the authors' time and effort spent on them.

The findings on the relation between feature variance and K-NN clustering may lead to a new research direction. Picking up certain features from generic pre-training for specific tasks is very interesting.

**Weaknesses:**

The paper sets ambitious goals by aiming to explore fundamental local behaviors of Vision Transformers (ViTs) and address significant questions. However, it falls short in several aspects, as outlined below:

1. Limited Novelty. While a pure experimental paper can definitely contribute to the research community, this paper's experimental settings lack innovation. The techniques employed, such as linear probing, K-NN for clustering, and the introduction of blur and noise for degradation, are rather basic. It's not that simplicity is a problem, but the experimental design should be more rigorous and well-founded, as discussed below.

2. Limited Evaluation Protocol. The paper aims to tackle four substantial questions, but it primarily relies on two experiments: clustering and retrieval. This limited evaluation protocol may not fully capture the models' true properties. For example, the clustering task involves only KNN and linear probing, which may not be suitable for the diversity of ViT features. There are alternative hypotheses, like multiple clusters in the features may correspond to the same semantic class (therefore KNN and linear probing may fail), or the features are not separable by linear methods (therefore linear probing may fail). To truly understand performance, fine-tuning the backbone and using a prediction head with better capacity should be conducted, which the paper lacks.

3. Limited Data: In addition to the limited evaluation protocol, the choice of data is also limited. For instance, the clustering experiments rely on Cityscapes data, which strongly biases towards scenes related to driving and certain classes like 'cars, trees, sky, road,' etc. Although the paper claims that findings generalize to ADE20k, the results are inconclusive with low accuracy (mostly around 5%, see Table 2 in the supplementary), making it difficult to draw meaningful conclusions. The same issue arises in the 'tracking' experiments, which are based on a subset of just four sequences from the BDD-100k dataset, another driving-related dataset.

4. Retrieval Experiment Disconnect: The retrieval experiments fail to directly address the question "At which granularity do transformers detect objects?" The experiments involve adding basic augmentations to assess embedding drift. However, the connection between these experiments and the underlying question is tenuous. The chosen augmentations are relatively simple and thus do not sufficiently represent real-world data variations. Essentially this experiment only tests the robustness of ViTs to certain handcrafted types of noise and blur.

5. Tracking Task Shortcomings: The paper's tracking tasks are not conducted using a proper tracking framework but rather rely on retrieval from different frames. This approach is suboptimal as matching between image frames may not be accurate. ViT features lack translational invariance, caused by positional embedding, so simple retrieval may fail if the object of interest moves. Consequently, this experiment only evaluates the consistency of ViT features across video frames and does not reflect the true tracking capabilities of ViTs.

In summary, while the paper's aspirations are commendable, it falls short in terms of experimental design, data choice, and providing insightful analyses, ultimately limiting its contributions to the research community.

**Questions:**

1. It's intriguing to investigate feature variance. Have you considered normalizing the features to have the same variance and then applying KNN?

2. Clustering at the patch level in a dense-prediction dataset seems questionable. In the case of Cityscapes, a single patch can contain multiple semantic classes, and assigning it to a specific class may not align with the desired behavior of ViTs. Have you explored alternative methods for this?

3. It would be good to extend the paper to more pre-training methods, including visual-language (CLIP) and segmentation (Segment Anything).

4. The paper presents numerous experiments but more in-depth discussions may be beneficial. For example, examining MAE feature variance is interesting. It would be good to show some visual results on how these high-variance features contribute to reconstruction.

5. Is there any reason why the tracking experiments are only conducted on 4 videos in BDD-100k? Is it possible to try more generic MOT datasets?

A minor point, in the introduction, relating the paper to NLP and LLM seems like hindsight, not a motivation.

---

### Official Review · Reviewer_HZLL · 2023-11-08

**Soundness:** 3 good
**Presentation:** 3 good
**Contribution:** 2 fair
**Rating:** 5
**Confidence:** 5

**Summary:**

This paper focus on local representative power of various self-supervised Vision Transformers (ViTs). Specifically, this paper presents a series of comparative analyses and corresponding conclusions through experiments in patch-level tasks such as few-shot semantic segmentation, instance identification, object retrieval and tracking.

**Strengths:**

- The authors have conducted a detailed analysis of the local representation capabilities of different pre-trained models, supported by numerous experiments. This includes five different pre-training ViTs and three tasks.

- The authors have provided detailed visualizations to aid in understanding the conclusions.

- The conclusions are stated clearly and concisely.

**Weaknesses:**

- This paper seems to lack a clear motivation, namely, what is the purpose of comparing and analyzing the local representational capabilities of different pre-trained models. It is necessary to explain where these conclusions can be applied or how they may inspire future research directions.

- The overall structure of this paper is somewhat loose. It is generally divided into sections by task, with each section offering some empirical observations. The authors are advised to try to organize their core conclusions and supplement experiments around those conclusions for more coherent organization.

- There are too many observations provided in this paper, but too little analysis behind them. For example, why is there such a big difference between MAE and SimMIM, both of which are masked image modeling methods, in the accuracy of the kNN middle layer (Figure 2) and robustness (Figure 3)? Especially for Figure 3, MAE seems to have a huge drop when adding larger disturbances, all below the performance of SimMIM. This should be explained.

- Section five of this paper seems to deliberately omit the analysis of SimMIM. Considering that the tasks are not very costly, the authors need to explain the specific reasons for this.

- The tasks presented in this paper are too simplistic, and given that current visual models cannot perform zero-shot or few-shot inference like large language models (LLMs), it is a matter to consider whether the analyses of these simplistic tasks can be transferred to the fine-tuning stage.

**Questions:**

Please refer to the weaknesses.